# GSN-HVNET: A Lightweight, Multi-Task Deep Learning Framework for Nuclei Segmentation and Classification

**DOI:** 10.3390/bioengineering10030393

**Published:** 2023-03-22

**Authors:** Tengfei Zhao, Chong Fu, Yunjia Tian, Wei Song, Chiu-Wing Sham

**Affiliations:** 1School of Computer Science and Engineering, Northeastern University, Shenyang 110819, China; 2Engineering Research Center of Security Technology of Complex Network System, Ministry of Education, Shenyang 110819, China; 3Key Laboratory of Intelligent Computing in Medical Image, Ministry of Education, Northeastern University, Shenyang 110819, China; 4State Grid Liaoning Information and Communication Company, Shenyang 110006, China; 5School of Computer Science, The University of Auckland, Auckland 1142, New Zealand

**Keywords:** joint nuclei segmentation and classification, lightweight, multi-task deep learning framework, Residual-Ghost-SN, Dense-Ghost-SN

## Abstract

Nuclei segmentation and classification are two basic and essential tasks in computer-aided diagnosis of digital pathology images, and those deep-learning-based methods have achieved significant success. Unfortunately, most of the existing studies accomplish the two tasks by splicing two related neural networks directly, resulting in repetitive computation efforts and a redundant-and-large neural network. Thus, this paper proposes a lightweight deep learning framework (GSN-HVNET) with an encoder–decoder structure for simultaneous segmentation and classification of nuclei. The decoder consists of three branches outputting the semantic segmentation of nuclei, the horizontal and vertical (HV) distances of nuclei pixels to their mass centers, and the class of each nucleus, respectively. The instance segmentation results are obtained by combing the outputs of the first and second branches. To reduce the computational cost and improve the network stability under small batch sizes, we propose two newly designed blocks, Residual-Ghost-SN (RGS) and Dense-Ghost-SN (DGS). Furthermore, considering the practical usage in pathological diagnosis, we redefine the classification principle of the CoNSeP dataset. Experimental results demonstrate that the proposed model outperforms other state-of-the-art models in terms of segmentation and classification accuracy by a significant margin while maintaining high computational efficiency.

## 1. Introduction

Over the past several years, deep-learning-based computer vision techniques have been extensively applied to computer-aided diagnosis (CAD). In computational pathology, pathological image analysis based on the deep learning method has proven powerful in improving efficiency and accuracy in cancer detection [1]. The morphology of the nuclei is the essential feature used by pathologists in cancer diagnosis and further cancer prognoses, such as predicting survival [2] and pathological grading of tumors [3]. Accurate nuclei segmentation and classification can advance the quality of tissue segmentation [4,5]. Nuclei segmentation is the crucial first step to obtaining the morphological features used in the downstream analysis. However, the morphological heterogeneity of nuclei makes studies challenging. The karyomorphism shows variability, while different diseases may cause chromatin abnormalities to exhibit variable size and shape patterns. Another severe problem is that the cells in a cancerous tumor are usually densely packed and even have more than one nucleus, causing overlapping nuclei. This overlapping brings difficulty for further research on separating neighboring instances via automatic segmentation.

Extracting each nucleus and distinguishing its type can promote the diagnostic potential in present-day digital pathology pipelines. For instance, precisely distinguishing each nucleus from tumors or lymphocytes can significantly facilitate downstream analysis of tumor-infiltrating lymphocytes (TIL), which has been proven effective in predicting cancer recurrence [6]. The nucleus-by-nucleus classification has become another problem researchers have been interested in recently due to the high variability and diversity of nuclei appearance in a whole slide image.

The current deep models for histopathology image diagnosis are mainly based on single-task learning. Single-task learning is designing a model for a specific task and then optimizing iteratively. In this case, the nuclei segmentation and classification tasks require two independent models, one for detecting the location of each nucleus and the other for classifying the type of nuclei [7,8]. For more complicated tasks, we are accustomed to modeling each part of the task by disassembling. However, there exists an obvious problem in this way. When modeling each sub-task, it is easy to ignore the relationships, conflicts, and constraints between different sub-tasks, resulting in the downgrading of the overall performance of the entire task.

To address the above issue, multi-task models have drawn much attention [9,10,11,12]. The multi-task models have the following advantages: (1) multiple tasks share the same model, reducing the amount of memory; (2) multiple tasks obtain results through a forward calculation at one time, and the inference speed increases; (3) associated tasks share information and complement each other, improving each tasks’ performance.

Recently, several multi-task deep models for histopathology image diagnosis have been suggested and achieved promising results [13,14,15]. Unfortunately, these approaches still suffer from efficiency issues, such as dealing with a cumbersome model with a huge amount of parameters. In addition, the classification on the CoNSeP dataset [13] seems hard to meet the needs of practical pathological diagnosis.

The present paper proposes a lightweight, multi-task deep learning framework for segmenting and classifying nuclei simultaneously. To address the problem of network stability encountered by batch normalization (BN) when dealing with small batch sizes, we introduce two newly designed blocks. We devise an efficient encoder-decoder architecture, where the encoder adopts our proposed RGS for down-sampling, while the decoder uses Dense-Ghost-Module (DGM) and convolution for up-sampling. By encoding the HV distance of nuclei pixels, we can obtain more representative features on the instance with fewer layers. Here, HV distance can be used to segment overlapping nuclei instances accurately. Later, the decoder using the output features of the encoder predicts nuclei types. According to the above characteristics, we call the proposed network GSN-HVNET. Our experimental results show that the proposed model can retain shallow features on nuclei to improve segmentation and classification accuracy. Our main contributions are outlined below:We propose a novel, lightweight, multi-task deep learning framework containing a unified model for segmentation and classification of nuclei instances simultaneously with superior efficiency and accuracy.We propose the newly designed RGS and DGS to improve accuracy and compress the training model.We redefine the classification principle of the CoNSeP dataset so that the auxiliary diagnostic results have practical significance in pathological diagnosis.Our experiments on the CoNSeP, Kumar, and CPM-17 datasets confirm the improvements to existing works [13,14]. Compared with the state-of-the-art HoVer-Net [13], the number of parameters is reduced by 64%. In addition, we try different batch sizes in our experiments and prove that batch size is no longer a strict limitation on the proposed network; even when a small batch is presented, the proposed network can maintain a high performance.

The remainder of this paper is organized as follows: Section 2 introduces the current research on applying learning algorithms in histopathology image analysis. Our new network architecture is presented in Section 3. We conduct experiments and show desirable results in Section 4. Finally, Section 5 concludes our work and gives a brief discussion of future work.

## 2. Related Work

### 2.1. Nuclei Segmentation

Nuclei segmentation is the crucial first step in computer-aided systems for cancer detection. Low level information analysis of histopathology images, such as histograms analysis [16,17,18,19,20], were often used for early nuclei segmentation algorithms. There was an obvious shortcoming that occurred to those algorithms. A certain threshold was hard to be determined to adapt to all scenarios. In [21], the authors proposed a fast and flexible segmentation algorithm based on computing the watersheds in digital grayscale images. Unfortunately, related experiments reported several false-positive segmentation cases. In [22], the authors proposed a novel, marker-controlled watershed based on mathematical morphology to segment clustered cells with less oversegmentation, designing a tracking method based on modified mean shift algorithm to segment undersegmented cells or merge oversegmented cells. In [23], the authors proposed a method combining region growing and machine learning to segment touching nuclei and classify them. In [24], the authors proposed an improved method, which used a joint optimization of a multiple-level set function to segment the cytoplasm and nuclei from clumps of overlapping cervical cells. In [25], the authors proposed using the graph theory technique to segment glands and computed a gland-score for estimating how similar a segmented region is to a gland. In [26], the authors proposed a superpixel-based segmentation technique with different morphological and clustering algorithms. Unfortunately, these existing segmentation algorithms cannot provide utterly reliable results because they need to manually extract nucleus features, which, thus, are inflexible and laborious to extend to a complex scenario.

Rather than manual feature extraction in traditional algorithms, deep learning methods can automatically extract a distinct set of features, and have been widely applied to nuclei segmentation [27]. For instance, U-Net has presented an outstanding performance in biomedical image segmentation [28]. In [29], the authors proposed a deep multi-scale neural network for accurately segmenting nuclei by improving sensitivity to hematoxylin intensity. In [30], to meet the challenge of segmenting overlapping or touching nuclei, the authors formulated the segmentation problem as a regression task of the distance map, and the nuclei boundary information was used as prior knowledge for a segmentation network. In [31], the authors proposed a contour-aware informative aggregation network with a multi-level information aggregation module between two task decoders: one of these segments the nuclei, and the other segments the contours.

### 2.2. Nuclei Classification

Nuclei classification is a vital step in histopathology image analysis, promoting downstream analysis such as evaluating cancer progression. Early studies utilize manually extracted features to classify the nuclei automatically. Typically, an SVM-based method [32] applied iterative feedback to obtain subtle and complex features of cellular morphology. Albeit showing good performance in high-penetrant phenotypes, it can hardly achieve a satisfying performance in lower-penetrant phenotypes. In [33], Ada-boost was used as the classifier to classify the nuclei after segmentation. The classifier was constructed based on intensity, texture, and morphology features. However, these machine learning methods manually extract features, and their representation ability and stability can still be affected by subjective factors to some extent.

Generally, a deep-learning-based nuclei classification model consists of two main phases. Firstly, each nucleus is segmented or detected using a deep model; then, those features are fed into a classifier to confirm nuclei types. For instance, in [34], the nuclei in colon cancer histology images were firstly detected using a spatially constrained CNN. Then, each nucleus with associated patches was fed into the convolution network to predict its type, i.e., inflammatory, healthy, or malignant epithelium.

## 3. Proposed Method

Figure 1 shows an overview of the GSN-HVNET for simultaneous nuclei instance segmentation and classification. The network input starts with 80×80×3 images, which are center patches cropped out from the sample images of size 270×270×3. The model can simultaneously segment the nuclei and predict nuclei types and HV-Maps (horizontal and vertical maps). After a post-processing procedure, the nuclei instance can be obtained using HV-Map and nuclei pixel predictions. The final output results can be obtained by combining the segmentation results with the nuclei-type predictions. In other words, the network can complete the segmentation and classification of nuclei instances at the last step.

### 3.1. Network Architecture

Figure 2 illustrates the detailed structure of the proposed GSN-HVNET. The proposed network consists of an encoder and a decoder for automatic segmentation and classification of nuclei instances. The encoder can extract an effective set of features; then, the output result of the encoder is used as the decoder input. The decoder contains three branches. Branch I (NSS) is used in nuclei semantic segmentation, and branch II (HV) predicts the HV distances of nuclei pixels to their mass centers. Nuclei types are predicted in branch III (NC). We combine the output of branch I and branch II to accomplish the instance segmentation. Then, the instance segmentation result combines the branch III output to accomplish automatic segmentation and classification of the nuclei instance. The encoder employs the proposed RGS, as discussed in Section 3.1.1. The details of GBS and RGS will be introduced in Section 3.1.2 and Section 3.1.3, respectively. In Section 3.1.4, the decoder designed with DGS will be described. The details of DGS will be presented in Section 3.1.5.

#### 3.1.1. Encoder

To extract a practical set of features, we design a novel residual ghost network as part of the encoder in the overall network. The network employs a Conv2D-SN-ReLU (CSR) and a series of 4 Residual-Ghost-Modules (RGMs) for down-sampling. Here, the CSR block is composed of a Conv2D, SN, and ReLU. An RGM consists of multiple instances of our improved Ghost-Block—Residual-Ghost-Block with switchable normalization (RGS) [35]. Benefiting from ghost convolution, our network requires much fewer parameters to generate abundant feature maps compared with using ordinary convolution, resulting in an improved computational efficiency of our encoder. Moreover, the SN can select an optimal combination of different normalizers for different normalization layers, improving the network stability, i.e., the accuracy is not affected by the batch size. Each RGM is used as a down-sampling level of 2, which means that the spatial resolution of the input is reduced by a factor of 2. We will give a detailed discussion on RGS in the two subsequent subsections.

#### 3.1.2. Ghost Block with Switchable Normalization

Figure 3 compares the structure of Ghost-Block-BN (GBB) [36] and our suggested Ghost-Block-SN (GBS). As is known, Ghost-Block can help a convolutional neural network to generate more features at a much lower cost. To do that, a Ghost-Block first generates several intrinsic feature maps using ordinary convolution operation and then uses cheap linear operations to expand the features and increase the channels. The computational cost of the linear operations on feature maps is much lower than traditional convolution and transcends other existing efficient works. We can customize the kernel size of the primary convolution in a Ghost-Block, and 1×1 point-wise convolution is employed in this paper. In the Residual-Ghost-Block (RGB), each Ghost-Block is followed by a BN layer, which offers several advantages, including stabilizing and speeding up the training procedure. However, the performance of GBB is severely restricted by the batch size. This is because BN only utilizes a single normalizer in the entire network, which can be unstable and hurt the accuracy in the case of a small batch size.

To solve the above problem, we apply switchable normalization (SN), which is robust to a wide range of batch sizes, whether a small batch size or not. As shown in Figure 4, SN measures channel-wise, layer-wise, and minibatch-wise statistics by using instance normalization (IN) [37], layer normalization (LN) [38], and batch normalization (BN) [39], respectively, and tries to find an optimal combination by learning their important weights, ensuring the stability and accuracy of the network in the case of small batch size.

#### 3.1.3. Residual Ghost Block with Switchable Normalization

Our RGS adopts the structure of residual block—the essential building unit of residual neural network (ResNet) [40]—owing to its outstanding performance. The key idea behind residual block is to reformulate the layers as learning residual functions with reference to the layer inputs, instead of learning unreferenced functions. As shown in Figure 5, we embed the proposed GBS in a residual block as RGS. Later, several RGSs are stacked to form the RGM. Our network contains of four stacked RGMs with 1, 2, 3, and 1 RGS, respectively. Compared with original ResNet-50, our network employs fewer building units to extract feature maps and reduce redundant features, leading to a reduction in model size. In addition, our proposed RGS is generic and can be used in the construction of other lightweight deep learning architectures.

#### 3.1.4. Decoder

As aforementioned, the decoder contains three branches to obtain accurate nuclei instance segmentation and classification simultaneously. These three branches adopt the same architecture consisting of a series of up-sampling operations and two Dense-Ghost-Modules. A DGM contains a series of cascading DGSs. Through stacking multiple DGSs, we can enrich the receptive field with relatively fewer parameters compared with the most popular Dense-Block, resulting in increased computational efficiency. As is known, low-level information is critical in segmentation tasks because it precisely helps to determine object boundaries. To make use of it, we adopt the skip connections to merge feature from each RGS in the encoder via the concatenation operation. The DGM follows the first and second up-sampling operations. There are eight and four DGSs in the first and second DGM, respectively. Each of the three branches contains three up-sampling steps, making the output feature the same dimension as the input image, i.e., 80×80×3. By combining the results of the two up-sampling branches, NSS and HV, we can obtain accurate boundaries of each individual cell nucleus, and thereby accomplish the nuclei instance segmentation. Compared with independent networks for different tasks, the proposed network is a unified model to simultaneously accomplish nuclei segmentation and classification, thus reducing the total training time.

#### 3.1.5. Dense Ghost Module with Switchable Normalization

In this part, we propose a novel module applied to the decoder of GSN-HVNET. An example of the proposed module is shown in Figure 6, in which n=4. Each DGS connects to other DGSs with forwarding feedback and employs GBS to extract feature maps. The feature maps from all preceding layers are utilized as current inputs, and the feature maps output by a DGS are used as inputs for all subsequent layers.

Thus, the proposed module can retain more abundant features as inputs of subsequent layers.

Similarly, benefiting from the lightweight nature of GBS, our proposed DGS utilizes fewer parameters to generate abundant feature maps and valid features compared with Dense-Block [41]. Moreover, it helps to avoid unnecessary calculations by reducing redundant feature maps. Particularly, the DGM can maintain its performance under a small mini-batch size.

#### 3.1.6. Joint Loss Function of GSN-HVNET

We design different loss functions for each different task. In Table 1, we define the notations for our works. The joint loss function $L_{Join}$ is defined by
(1)$$L_{Join}=L_{NSS}+L_{HV}+L_{NC}.$$

The NSS branch corresponds to a semantic segmentation task, and its loss function is designed using BCE loss and dice loss. It is defined by
(2)$$L_{NSS}=\lambda_{a}L_{BCE}+\lambda_{b}L_{DICE},$$
where $L_{BCE}$ and $L_{DICE}$ represent the binary cross-entropy loss function and dice loss function for the output of the NSS branch, respectively. The $\lambda_{a}$ and $\lambda_{b}$ are scalars that give weights to their associated loss function. The above two functions are defined by
(3)$$L_{BCE}=-\left[\frac{1}{n}\sum_{i=1}^{N}\sum_{k=1}^{K}X_{i,k}\left(I\right)\log Y_{i,k}\left(I\right)+\sum_{i=1}^{N}\sum_{k=1}^{K}\left(1-X_{i,k}\left(I\right)\right)\log\left(1-Y_{i,k}\left(I\right)\right)\right]$$
and
(4)$$L_{DICE}=1-\frac{2\times \sum_{i=1}^{N}\left(Y_{i}\left(I\right)\times X_{i}\left(I\right)\right)+\epsilon }{\sum_{i=1}^{N}Y_{i}\left(I\right)+\sum_{i=1}^{N}X_{i}\left(I\right)+\epsilon },$$
where *X* represents the ground truth, *Y* denotes the prediction, and *K* represents the number of categories. In order to avoid zero denominators, we set ϵ to $1.0\times e^{-4}$ in the numerator and denominator.

The loss function for the HV branch is defined by
(5)$$L_{HV}=\lambda_{c}L_{MSE}+\lambda_{d}L_{MSGE},$$
where $L_{MSE}$ represents the mean squared error measuring the difference between the HV distances prediction and the ground truth, $\lambda_{c}$ and $\lambda_{d}$ are the weights of their associated loss function. The loss function $L_{MSGE}$ is used to calculate the gradients of the mean squared error between HV maps and ground truth. $L_{MSE}$ and $L_{MSGE}$ are defined by
(6)$$L_{MSE}=\frac{1}{n}\sum_{n=1}^{n}{\left(p_{i}\left(I\right)-\Gamma_{i}\left(I\right)\right)}^{2}$$
and
(7)$$L_{MSGE}=\frac{1}{m}\sum_{i\in M}{\left(\nabla_{x}\left(p_{i,x}\left(I\right)-\Gamma_{i,x}\left(I\right)\right)\right)}^{2}+\frac{1}{m}\sum_{i\in M}{\left(\nabla_{y}\left(p_{i,y}\left(I\right)-\Gamma_{i,y}\left(I\right)\right)\right)}^{2},$$
where *I* represents the input image and $p_{i}\left(I\right)$ is defined as the regression output of HV branch. The pixel-wise softmax predictions of NSS and NC branches are represented by $q_{i}\left(I\right)$ and $r_{i}\left(I\right)$, respectively. $\Gamma_{i}\left(I\right)$ denotes the ground truth of the HV distance of nuclei pixels to their mass centers.

The loss function of $L_{NC}$ is defined by
(8)$$L_{NC}=\lambda_{e}L_{BCE}+\lambda_{f}L_{DICE}.$$

Similarly, $\lambda_{e}$ and $\lambda_{f}$ are used to balance the two loss functions $L_{BCE}$ and $L_{DICE}$.

### 3.2. Post-Processing

The proposed network produces three outputs. To obtain the nuclei location and separate overlapping or clustered nuclei, we need to post-process the output of NSS and HV. Within each HV map, there are significant differences between pixels in adjacent instances. Using this property, we can calculate the gradient so as to separate the clustered nuclei. To do that, we have
(9)$$S_{m}=max\left(Hor\left(p_{hor}\right),Ver\left(p_{ver}\right)\right),$$
where $p_{hor}$ and $p_{ver}$ represent the horizontal and vertical predictions produced by the HV branch, and Hor and Ver refer to the horizontal and vertical components, respectively, of the Sobel operator, which calculates the horizontal and vertical derivative approximations. In Figure 1, $S_{m}$ highlights the regions where pixels in adjacent regions of two instances differ significantly in the horizontal and vertical maps.

We compute the marker *M* according to
(10)$$M=\sigma (\tau \left(q,h\right)-\tau \left(S_{m},k\right)),$$
where *q* is the output probability map of the NSS branch and τ(q,h) is a threshold function acting on *q* and sets values above *h* to 1 or 0; otherwise, σ is a rectifier setting all negative value to 0 and M is the output marker. We can obtain desired segmentation results by choosing appropriate *h* and *k*.

Next, we compute the energy landscape *E* according to
(11)$$E=\left[1-\tau \left(S_{m},k\right)\right]*\tau \left(q,h\right).$$

Finally, given the energy landscape *E*, a marker-controlled watershed is carried out using *M* as the marker to determine how to split τ(q,h), given the energy landscape *E*. The task of joint segmentation and classification of nuclei requires converting per-pixel nuclei type prediction in the NSS branch to the prediction of the type of nuclei instances. To do that, we combine the post-processing result with NC branch output.

## 4. Experiment

### 4.1. Datasets and Implementation

In our experiment, we adopt three authoritative nuclei datasets: CoNSeP [13], Kumar [42], and CPM-17 [43]. Table 2 describes these datasets used in our experiment. The CoNSeP dataset, extracted from 16 colorectal adenocarcinoma (CRA) WSIs, consists of 41 hematoxylineosin (H&E) staining images, each of size 1000×1000 at 40× objective magnification. In CoNSeP dataset, tumor regions, stroma, muscular, fat, glandular, and collagen can be observed. In addition to containing different tissue components, seven nuclei types are provided, including malignant/dysplastic epithelial nuclei, normal epithelium, inflammatory, fibroblast, muscle, endothelial, and miscellaneous. In [13], the authors combined the original seven categories into four categories, of which malignant/dysplastic epithelial and normal epithelial were combined into a single type corresponding to the epithelial class, and fibroblast, muscle, and endothelial were combined into a single type corresponding to the spindle-shaped class. However, in practical clinical diagnosis, a CAD system should mainly focus on the identification of lesion area. To address this issue, we reclassified this dataset in our experiment. Specially, the normal epithelium, fibroblast, muscle, endothelial, and miscellaneous were combined into a single type corresponding to normal region, and the malignant/dysplastic epithelial and inflammatory are considered as two separate types—that is, the reclassified contain three nuclei categories as well as the background category. With this classification rule, our model can directly report the types of nuclei in lesion areas.

Kumar is an annotated dataset containing over 13,000 segmented nuclei from four different organs—breast, kidney, liver, and prostate—of 16 patients. The CPM-17 dataset provides the tissue image with labels for nuclei segmentation and classification. It is obtained from patients with head and neck squamous cell (HNSCC), glioblastoma multiforme (GBM), non-small cell lung cancer (NSCLC), and lower-grade glioma tumors (LGG). Some examples taken from these datasets are shown in Figure 7.

We run our code on a server equipped with an NVIDIA Geforce RTX 3090 GPU and Intel(R) Xeon(R) Gold 5118 CPU. During the training phase, we performed data augmentation to augment the training data. We randomly combined zooming, channel shifting, shearing, rotating, and horizontal/vertical flipping, which cropped the original image into 270 × 270 sub-images. We used Kaiming normalization [44] to initialize weights and set initial bias as false. We used Adam [45] as the optimizer, with a trainable batch size of 4. We set an initial learning rate as $1.0\times e^{-4}$ and weight decay as 0.1. The six hyper-parameters $\lambda_{a}$, $\lambda_{b}$, $\lambda_{c}$, $\lambda_{d}$, $\lambda_{e}$, and $\lambda_{f}$ used for balancing the joint loss function are tuned to be {1,1,1,1,2,1} on the validation set.

### 4.2. Evaluation Metrics

#### 4.2.1. Nuclei Instance Segmentation Evaluation

The segmentation of the nuclei instances can be divided into three sub-tasks; these three sub-tasks are the separation of the nuclei from the background, the detection of individual nuclei instances, and the segmentation of each detected instance. The Ensemble Dice [43] and Aggregated Jaccard Index [42] are two popular metrics used to measure the performance of nuclei instance segmentation. To better investigate the proposed method, we need to measure the performance of each sub-task. The dice coefficient ($F_{1}$ score) is defined by
(12)$$Dice\_coef=\frac{\left|TP\right|}{\left|TP\right|+\frac{1}{2}\left|FP\right|+\frac{1}{2}\left|FN\right|}=\frac{2\times (X\cap Y)}{\left(|X|+|Y|\right)},$$
where TP represents the true-positive rate, FP represents the false-positive rate, and FN represents the false-negative rate. *X* and *Y* represent the ground truth and prediction, respectively.

The AJI calculates the ratio of an aggregated intersection cardinality to an aggregated union cardinality between the ground truth and prediction. It is defined by
(13)$$AJI=\frac{\sum_{i=1}^{N}\left|G_{i}\cap P_{M}^{i}\right|}{\sum_{i=1}^{N}\left|G_{i}\cup P_{M}^{i}\right|+\sum_{F\in U}\left|P_{F}\right|},$$
where $G_{i}$ is the *i*th nucleus from the ground truth with N nuclei. $P_{M}^{i}$ represents the *M*th connected component in prediction, which has the largest Jaccard Index with $G_{i}$, and where each *M* cannot be utilized more than once. *U* is a set representing the connected component in the prediction without the corresponding ground truth.

Unfortunately, $F_{1}$ score and AJI only calculate an overall score for the instance segmentation quality. In addition, the two metrics suffer from a limitation that they will produce excessive penalization and result in an abnormal score for overlapping regions.

To take a measurement of each sub-task, we take advantage of panoptic quality [46] with accurate quantification and interpretability to measure the performance of nuclei instance segmentation. The panoptic quality for nuclei instance segmentation is defined by
(14)$$PQ=DQ\times SQ=Dice\_coef\times \frac{\sum_{(x,y)\in TP}IoU\left(x,y\right)}{\left|TP\right|},$$
where *x* and *y* denote a ground truth component and a prediction component, respectively. The IoU represents the intersection over union. Each (x,y) must be unique over the whole set of prediction and ground truth segments, if their IoU(x,y)>0.5. DQ and SQ help to give a direct insight into detecting individual nuclear instances and segmenting each detected instance. Therefore, PQ can serve as the objective evaluation criteria for measuring the performance of the nuclei instance segmentation task.

To demonstrate the effectiveness of the proposed method, we use the following three metrics. Dice coefficient and PQ are used to measure the separation of all nuclei from the background and serve as a unified score for comparison, respectively. The AJI is used for the comparison with other methods. In this study, these three metrics serve as objective evaluation criteria. As the most reliable assessment of the segmentation quality, the subjective evaluation can also be carried out in practical applications.

#### 4.2.2. Nuclei Classification Evaluation

Nuclei classification is influenced by nuclei instance segmentation. The whole measurement for nuclei type classification should include nuclei instance segmentation. HoVer-Net [13] defines an efficient evaluation, which is defined by
(15)$$F_{c}^{t}=\frac{2(TP_{c}+TNc)}{2\left(TP_{c}+TN_{c}\right)+2\left(FP_{c}+FN_{c}\right)+\left(FP_{d}+FN_{d}\right)},$$
where $FP_{d}$ and $FN_{d}$ are false-positive and false-negative in detecting ground truth instances, respectively. $TP_{c}$, $TN_{c}$, $FP_{c}$, and $FN_{c}$ denote true-positive, true-negative, false-positive, and false-negative, respectively.

### 4.3. Experimental Results

Table 3 compares the number of trainable parameters of the proposed and other popular models. As can be seen from this table, our model gives the smallest size among all others in terms of the nuclei segmentation task and the second smallest size in term of the joint segmentation and classification task. Consequently, our model offers a high degree of computational efficiency. Table 4 compares the Dice scores of the proposed and two state-of-the-art models working with small mini-batch sizes. The results indicate that the proposed model appears more stable, i.e., our model can work well on small-memory-capacity GPUs, such as the NVIDIA 1080ti or 2080ti, thus reducing the hardware cost. The model size is significantly smaller than other compared networks.

The proposed network is measured by the three kinds of metrics discussed above, compared with baselines and other state-of-the-art networks, and the results are reported in Table 5. The results indicate that our proposed network achieves the highest accuracy among all the others. Moreover, even though the DIST model has fewer parameters than ours on joint nuclei segmentation classification task, its segmentation performance is worse than ours by a large margin on all three datasets. Therefore, our network offers an optimal trade-off between accuracy and efficiency.

As aforementioned, the 4-class nuclei classification carried out in HoVer-Net is impractical for use in practical pathological diagnosis. Accordingly, we have reclassified the data. Table 6 lists the comparative results for 3-class nuclei classification on the CoNSeP dataset. Here, $F_{d}$ denotes the $F_{1}$ score for nuclei detection. $F_{c}^{1}$, $F_{c}^{2}$, and $F_{c}^{3}$ denote the classification score for healthy, inflammatory, and malignant/dysplastic epithelium classes, respectively. The results show that the proposed network outperforms all the others in terms of $F_{c}^{1}$, $F_{c}^{2}$, and $F_{c}^{3}$ scores. In Figure 8, we illustrate the results of nuclei segmentation and classification on the sample images and compare them with those of [13,14,15,30].

As can be seen from this figure, our lightweight method is successful in segmenting overlapping and clustered nuclei. It is also excellent to complete the task of nuclei classification at the same time. Overall, our proposed model achieves state-of-the-art accuracy on nuclei segmentation and classification tasks while maintaining low computation cost. Our idea can be directly deployed in the cell pathology diagnosis system to reduce the workload of pathologists.

## 5. Conclusions

In this paper, we designed a lightweight, multi-task deep learning framework for nuclei segmentation and classification. Our model follows an encoder-decoder architecture, and the decoder consists of three branches, each outputting a prediction for a sub-task. To sufficiently use the correlation among the three branches, we employ NSS and HV branches to complete the nuclei instance segmentation and use NC branch to predict the classes of each nucleus in a learning process. Two newly designed blocks, Residual-Ghost-SN and Dense-Ghost-SN, are employed in the encoder and decoder parts, respectively, to reduce the computational cost and improve the network stability under small batch sizes. Extensive experiments have been carried out on the CoNSeP, Kumar, and CPM-17 datasets, and the results demonstrate that our model offers a state-of-the-art trade-off between computational efficiency and both segmentation and classification accuracy.

Ultimately, our idea is generic, and can be easily deployed to other histopathology images analysis works. Moreover, the blocks proposed in this paper, including Residual-Ghost-SN and Dense-Ghost-SN, are also generic and can be flexibly embedded into other deep CNNs for histopathology image diagnostic tasks. However, regarding their application in the field of natural images, we have not conducted experiments, and the effects cannot be guaranteed. Thus, we pose this as an open problem and expect to provide a theoretical analysis with complete proof in further research.

## Figures and Tables

**Figure 1 bioengineering-10-00393-f001:**
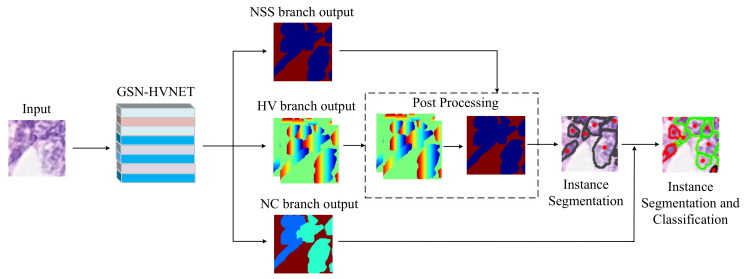
An overview of the GSN-HVNET for simultaneous nuclei instance segmentation and classification. The NSS branch achieves nuclei semantic segmentation, and the HV branch predicts the HV distances of nuclei pixels to their mass centers. Nuclei types are predicted in the NC branch. The nuclei instance segmentation can be accomplished by combining the output of the NSS and HV branches.

**Figure 2 bioengineering-10-00393-f002:**
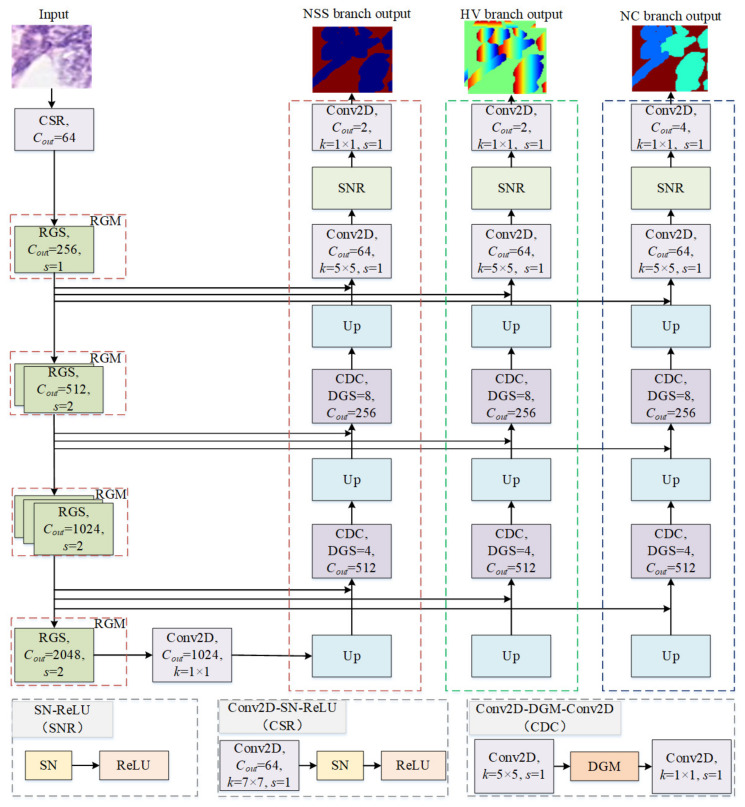
The structure of GSN-HVNET. Our proposed network contains an encoder and a decoder. The encoder, which extracts an effective set of features, is composed of a CSR block, four RGMs, and a Conv2D. The decoder is composed of three branches to achieve accurate nuclei instance segmentation and classification simultaneously.

**Figure 3 bioengineering-10-00393-f003:**
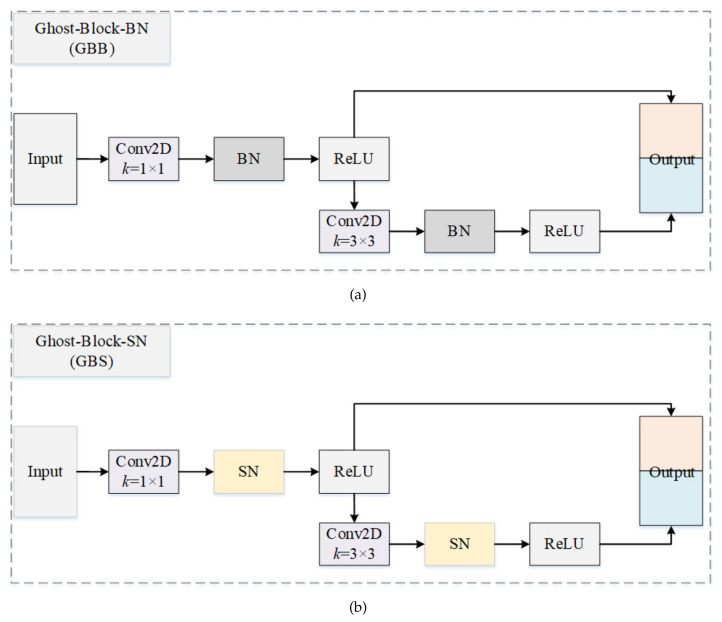
An illustration of the ghost block and the improved ghost block with switchable normalization. (**a**) Ghost block with batch normalization. (**b**) Ghost block with switchable normalization.

**Figure 4 bioengineering-10-00393-f004:**
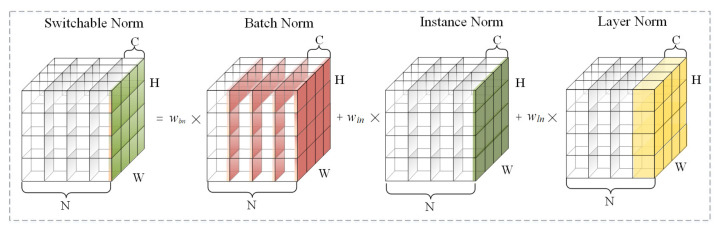
Switchable normalization. It learns to select different normalizers for different normalization layers of a deep neural network.

**Figure 5 bioengineering-10-00393-f005:**
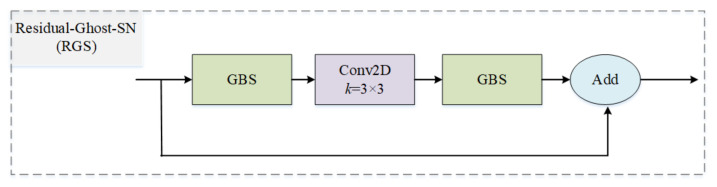
Residual ghost block with switchable normalization. The GBS denotes the ghost block with switchable normalization.

**Figure 6 bioengineering-10-00393-f006:**
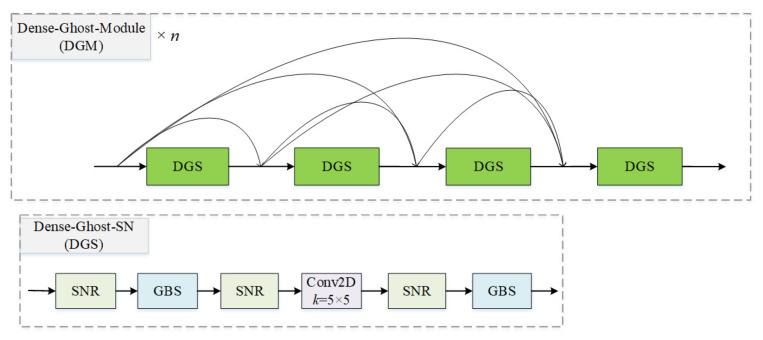
Dense ghost module with switchable normalization. The GBS and SNR denote the ghost block with switchable normalization and switchable normalization with ReLU, respectively.

**Figure 7 bioengineering-10-00393-f007:**
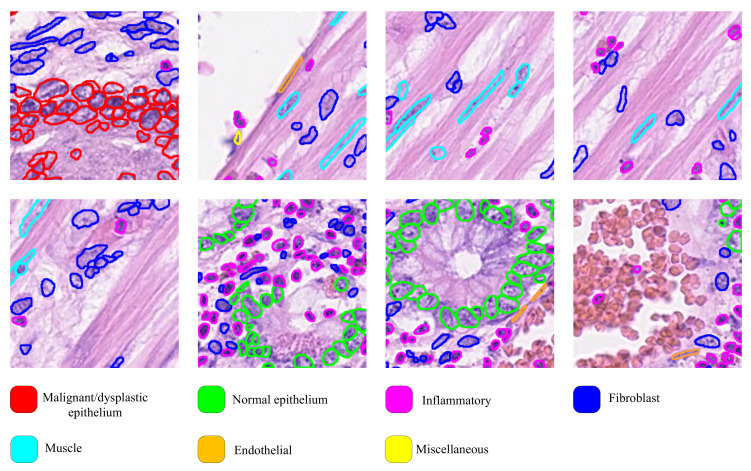
The sample clipping region is extracted from the CoNSeP dataset, where the color of each nuclear boundary indicates its category.

**Figure 8 bioengineering-10-00393-f008:**
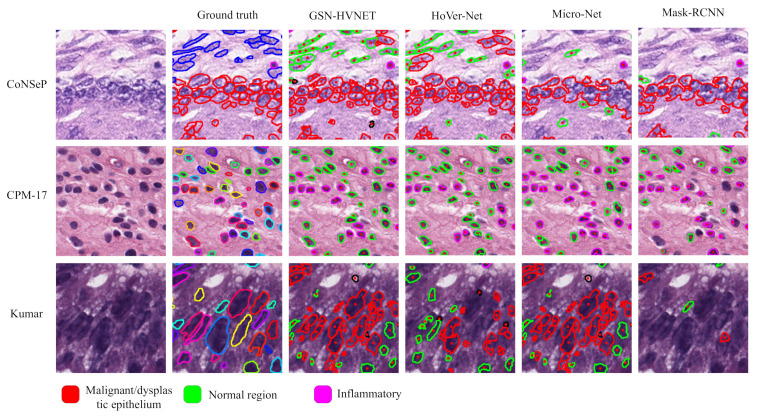
Comparative results for nuclei classification and segmentation. The normal epithelium, fibroblast, muscle, endothelial, and miscellaneous are combined into a single type corresponding to the normal region, and the malignant/dysplastic epithelial and inflammatory are considered as two separate types.

**Table 1 bioengineering-10-00393-t001:** The definition of notations.

Notation	Definition
$h_{ncij}$	The value of a pixel before normalization.
$\hat{h_{ncij}}$,	The value of a pixel after normalization.
γ, β	Scale and shift parameter
$I_{k}$, $\left|I_{k}\right|$	A set of pixels, and the number of pixels in $I_{k}$.
*L*, λ	*L* denotes the loss function and λ represents its parameters.
*I*	The input image.
$\Gamma_{i}\left(I\right)$	The HV distance of nuclei pixels to their mass centers.
$p_{i}\left(I\right)$	The regression output of HV branch.
$q_{i}\left(I\right)$	The pixel-wise and softmax predictions of NSS branch.
$r_{i}\left(I\right)$	The pixel-wise and softmax predictions of NC branch.
*E*	The energy landspace.
$F_{c}^{t}$	The whole measurement for nuclei type classification and nuclei instance segmentation.
FP, FN	False-positive, false-negative.
TP, TN	True-positive, true-negative.

**Table 2 bioengineering-10-00393-t002:** Description of the dataset used in our experiment. The Seg denotes the dataset with segmentation labels and the Class denotes the dataset with classification labels.

	CoNSeP	CPM-17	Kumar
Total numbers of nuclei	24,319	7570	21,623
Labeled nuclei	24,319	0	0
Number of images	41	32	30
Origin	UHCW	TCGA	TCGA
Magnification	40×	40× & 20 ×	40×
Size of images	1000 × 1000	500 × 500 to 600 × 600	1000 × 1000
Seg/Class	Seg&Class	Seg	Seg
Number of cancer types	1	4	8

**Table 3 bioengineering-10-00393-t003:** Comparative results for the number of trainable parameters of different networks for nuclei segmentation and classification. The Seg denotes the single-task network for segmentation. The Seg&Class denotes the multi-tasking network for simultaneous segmentation and classification.

Method	Seg/Class	Parameters
HoVer-Net [13]	Seg	42.94M
HoVer-Net [13]	Seg&Class	52.20M
Micro-Net [14]	Seg&Class	183.67M
DIST [30]	Seg&Class	8.81M
DCAN [47]	Seg	39.54M
SegNet [48]	Seg	28.07M
FCN8 [49]	Seg	128.05M
U-Net [28]	Seg&Class	35.23M
Mask-RCNN [15]	Seg&Class	44.17M
Our proposed	Seg	15.03M
Our proposed	Seg&Class	32.52M

**Table 4 bioengineering-10-00393-t004:** Comparative results for different mini-batch sizes presenting in three multi-tasking networks. The Dice coefficient is used to evaluate the segmentation performance on the CoNSeP, Kumar, and CPM-17 datasets.

Batch Size	Our Proposed			HoVer-Net			Micro-Net		
	Dice			Dice			Dice		
	CoNSeP	Kumar	CPM-17	CoNSeP	Kumar	CPM-17	CoNSeP	Kumar	CPM-17
1	0.821	0.851	0.865	0.816	0.794	0.843	0.752	0.759	0.828
2	0.830	0.844	0.870	0.806	0.804	0.875	0.764	0.785	0.857
3	0.839	0.842	0.870	0.835	0.819	0.879	0.758	0.794	0.859

**Table 5 bioengineering-10-00393-t005:** Comparative results for nuclei segmentation. The Dice coefficient, AJI, and PQ are used to evaluate the instance segmentation performance of ten networks on the CoNSeP, Kumar, and CPM-17 datasets.

Method	CoNSeP			Kumar			CPM-17		
	Dice	AJI	PQ	Dice	AJI	PQ	Dice	AJI	PQ
HoVer-Net [13]	0.838	0.525	0.494	0.826	0.618	0.597	0.869	0.705	0.697
SegNet [48]	0.796	0.194	0.270	0.811	0.377	0.407	0.857	0.491	0.531
FCN8 [49]	0.756	0.123	0.163	0.797	0.281	0.312	0.840	0.397	0.435
U-Net [28]	0.724	0.482	0.328	0.758	0.556	0.478	0.813	0.643	0.578
DIST [30]	0.798	0.495	0.386	0.789	0.559	0.443	0.826	0.616	0.504
DCAN [47]	0.733	0.289	0.256	0.792	0.525	0.492	0.828	0.561	0.545
Micro-Net [14]	0.784	0.518	0.421	0.797	0.560	0.519	0.857	0.668	0.661
Mask-RCNN [15]	0.740	0.474	0.460	0.760	0.546	0.509	0.850	0.684	0.674
CIA-Net [31]	-	-	-	0.818	0.620	0.577	-	-	-
Our proposed	0.861	0.602	0.566	0.879	0.635	0.644	0.899	0.701	0.683

**Table 6 bioengineering-10-00393-t006:** Comparative results for 3-class nuclei classification on the CoNSeP dataset. $F_{d}$ denotes the $F_{1}$ score for nuclei detection. $F_{c}^{1}$, $F_{c}^{2}$, and $F_{c}^{3}$ denote the classification score for healthy, inflammatory, and malignant/dysplastic epithelium classes, respectively.

Method	$F_{d}$	$F_{c}^{1}$	$F_{c}^{2}$	$F_{c}^{3}$
HoVer-Net [13]	0.784	0.488	0.525	0.517
Micro-Net [14]	0.812	0.487	0.549	0.546
DIST [30]	0.782	0.489	0.569	0.526
Mask-RCNN [15]	0.701	0.413	0.568	0.514
Our proposed	0.820	0.514	0.572	0.519

## Data Availability

The data that support the findings of this study are available from the corresponding author upon reasonable request.

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
