# Peer review of "GSN-HVNET: A Lightweight, Multi-Task Deep Learning Framework for Nuclei Segmentation and Classification"

_bioengineering, 2023, doi:10.3390/bioengineering10030393_

Round 1

Reviewer 1 Report

The paper discusses nuclei segmentation and classification challenges and presents a multi-task-based deep learning method, GSN-HVNET. The authors compared the performance of the proposed method with other previous works using three different datasets.

I enjoyed reading this paper. The motivation of the problem is clear, and the presentation of the proposed method is great. The performance results are impressive for using many other data sets and methods. The only comment I want to give is about the caption of Figure 2. It would be nice to see more descriptions in the caption. Also, it is hard to find the details of the CSR block in the figure and the text. 

Author Response

First we would like to thank the reviewers for their valuable comments and effort to improve the manuscript. In the following we outline each change made (point by point) as raised in the reviewers’ comments.

Reviewer 2 Report

On the basis of the insufficiency of the previous deep learning model in computer aided diagnosis of digital pathology images, the authors proposed a lightweight deep learning framework (GSN-5HVNET) with an encoder-decoder structure for simultaneous segmentation and classification of nuclei. Their results show better performance in segmentation and classification accuracy. Thus this work is recommended to accept after minor revision.

1. The authors should more details that verify their model.

Author Response

(The authors gave the same response as above.)
